# Preliminary brain-behavioral neural correlates of anterior cruciate ligament injury risk landing biomechanics using a novel bilateral leg press neuroimaging paradigm

Dustin R. Grooms[1,2,3]*, Jed A. Diekfuss[4,5,6], Cody R. Criss[2], Manish Anand[4,5,6,7], Alexis B. Slutsky-Ganesh[4,5,6,8], Christopher A. DiCesare[9], Gregory D. Myer[4,5,6,10]

1 School of Applied Health Sciences and Wellness, College of Health Sciences and Professions, Ohio University, Athens, OH, United States of America, 2 Ohio Musculoskeletal & Neurological Institute, Ohio University, Athens, OH, United States of America, 3 Division of Physical Therapy, School of Rehabilitation and Communication Sciences, College of Health Sciences and Professions, Ohio University, Athens, OH, United States of America, 4 Emory Sports Performance and Research Center, Flowery Branch, GA, United States of America, 5 Emory Sports Medicine Center, Atlanta, GA, United States of America, 6 Department of Orthopaedics, Emory University School of Medicine, Atlanta, GA, United States of America, 7 Department of Mechanical Engineering, Indian Institute of Technology Madras, Chennai, TN, India, 8 Department of Kinesiology, University of North Carolina at Greensboro, Greensboro, NC, United States of America, 9 Department of Mechanical Engineering, University of Michigan, Ann Arbor, MI, United States of America, 10 The Micheli Center for Sports Injury Prevention, Waltham, MA, United States of America

* groomsd@ohio.edu

**Editor:** Riccardo Di Giminiani, University of L'Aquila Department of Clinical Sciences and Applied Biotechnology: Universita degli Studi dell'Aquila Dipartimento di Scienze Cliniche Applicate e Biotecnologiche, ITALY

## Abstract

Anterior cruciate ligament (ACL) injury risk reduction strategies primarily focus on biomechanical factors related to frontal plane knee motion and loading. Although central nervous system processing has emerged as a contributor to injury risk, brain activity associated with the resultant ACL injury-risk biomechanics is limited. Thus, the purposes of this preliminary study were to determine the relationship between bilateral motor control brain activity and injury risk biomechanics and isolate differences in brain activity for those who demonstrate high versus low ACL injury risk. Thirty-one high school female athletes completed a novel, multi-joint leg press during brain functional magnetic resonance imaging (fMRI) to characterize bilateral motor control brain activity. Athletes also completed an established biomechanical assessment of ACL injury risk biomechanics within a 3D motion analysis laboratory. Knee abduction moments during landing were modelled as a covariate of interest within the fMRI analyses to identify directional relationships with brain activity and an injury-risk group classification analysis, based on established knee abduction moment cut-points. Greater landing knee abduction moments were associated with greater lingual gyrus, intra-calcarine cortex, posterior cingulate cortex and precuneus activity when performing the bilateral leg press (all $z > 3.1$, $p < .05$; multiple comparison corrected). In the follow-up injury-risk classification analysis, those classified as high ACL injury-risk had greater activity in the lingual gyrus, parietal cortex and bilateral primary and secondary motor cortices relative to those classified as low ACL injury-risk (all $z > 3.1$, $p < .05$; multiple comparison corrected). In young female athletes, elevated brain activity for bilateral leg motor control in regions that integrate sensory, spatial, and attentional information were related to ACL

**Data Availability Statement:** ***PA @ ACCEPT: Please request repository info at accept*** The data underlying the primary results are provided in supplement spreadsheet, however spreadsheets do not capture the full context of the neuroimaging data. The authors will share any further data upon request. Upon completion of study activities the raw data will be held in a public repository without restrictions.

**Funding:** This work was supported by NIH/NIAMS R01AR076153, R01AR077248 & U01AR067997 & US Department of Defense CDMRP award 81XWH-18-1-0707. Funders played no role in study design, data collection, analysis, or decision to publish or manuscript preparation. Opinions, interpretations, conclusions, and recommendations are those of the author and are not necessarily endorsed by the Department of Defense or the National Institutes of Health.

**Competing interests:** DRG has current and ongoing funding support from the National Institutes of Health/National Institute of Arthritis and Musculoskeletal and Skin Diseases (R01AR076153, R01AR077248) and the US Department of Defense Congressionally Directed Medical Research Program Peer Reviewed Orthopaedic Research Program (OR170266), research award (81XWH-18-1-0707). GDM consults with Commercial entities to support application to the US Food and Drug Administration but has no financial interest in the commercialization of the products. GDM's institution receives current and ongoing grant funding from National Institutes of Health/NIAMS Grants U01AR067997, R01 AR070474, R01AR055563, R01AR076153, R01 AR077248 and has received industry sponsored research funding related to brain injury prevention and assessment with Q30 Innovations, LLC, and ElMinda, Ltd. Dr. Myer receives author royalties from Human Kinetics and Wolters Kluwer. Dr. Myer is an inventor of biofeedback technologies (Patent No: US11350854B2, Augmented and Virtual reality for Sport Performance and Injury Prevention Application, Approval Date: 06/07/2022, Software Copyrighted.) designed to enhance rehabilitation and prevent injuries and receives licensing royalties. This does not alter our adherence to PLOS ONE policies on sharing data and materials.

injury-risk landing biomechanics. These data implicate crossmodal visual and proprioceptive integration brain activity and knee spatial awareness as potential neurotherapeutic targets to optimize ACL injury-risk reduction strategies.

## Introduction

Anterior cruciate ligament (ACL) rupture is a common sport- or physical activity-related injury among adolescent and young athletes, causing knee instability, loss of function, and increased risk of rapid early-onset osteoarthritis [1, 2]. Biomechanical injury risk factors include asymmetrical ground reaction forces, excessive hip internal rotation and knee abduction moments, and decreased knee and hip flexion [3, 4]. These biomechanical considerations have guided neuromuscular injury-reduction training protocols to employ a combination of strengthening, plyometrics, and movement control exercises to improve an athlete's motor coordination, specifically to limit knee frontal plane motion and loading [5]. A recent summary meta-analysis indicated these programs are capable of reducing ACL injury-risk, but the effectiveness of such programs could be improved as they require nearly 100 patients to be treated to prevent one ACL injury and ~50% of injury risk variance unexplained [6–8]. Thus, despite the moderate efficacy of ACL injury reduction programs, ACL injury rates have continued to increase over the past decade [9], further indicating contemporary programs have not reached their full potential.

Recent reports indicate a missing component is the failure to consider the underlying brain activity that results in the motor coordination patterns that increase injury risk [10, 11]. The failure to systematically target neural mechanistic drivers of motor function and control has plausibly limited neuromuscular training efficacy, requiring a high dosage for effect and with limited retention of injury-resistant neuromuscular control [12]. Numerous studies have thus aimed to isolate how central nervous system functioning (i.e., in the brain and spinal cord) contributes to ACL injury [13]. Specifically, central nervous system activity and connectivity for movement—not just physical factors in isolation (strength, biomechanics)—may explain why the majority of ACL injuries occur via non-contact mechanisms as a result of motor coordination errors while navigating through the athletic field [11, 14, 15]. The sensory-motor coordination error nature of injury having a potential neural basis is supported by alterations in brain functional connectivity between regions important for sensorimotor control [10, 16] and quadriceps neural inhibition [17] being prospectively identified for athletes who subsequently experienced an ACL injury. Further, athletes who presented with high ACL injury risk landing biomechanics exhibited more deterministic, and potentially maladaptive, patterns of electrocortical activity within frequency bands important for attention, cognition, and sensorimotor control [18]. However, the CNS assessments from these studies were completed while the participant was at rest (resting-state fMRI and EEG, respectively), warranting approaches that can assess brain activity during active, lower extremity movements. While historically constrained by technological and analytical limitations (e.g., management of head motion artifact), novel neuroimaging methods of unilateral lower extremity movement have been implemented successfully to assay knee motor brain activity for ACL deficient and reconstructed patients (e.g., flexion/extension [19, 20], hip-knee movement [21], joint position [22] and force control [23]). Prior neuroimaging investigations have further employed bilateral movements that simulate gait and pedaling in healthy and pathological populations [24, 25], but are limited in generalizability to ACL injury risk as they do not allow for simultaneous multiplanar

knee excursion as synonymous with ACL injury events and clinical neuromuscular control assessment [4, 26–30].

As ACL injury screening and neuromuscular training have adopted squat-like movements due to association with injury [31] and high correlation with landing mechanics [28, 32], quantifying the associated underlying brain activation of such bilaterally coordinated movement could identify unique neurological factors contributing to ACL injury risk movement coordination. Specifically, by isolating the neural mechanisms underlying excessive frontal plane motion loading that contributes to ACL injury-risk biomechanics, interventions could target precise neural processes to optimize the retention and transfer of injury-resistant neuromuscular control. Thus, the purposes of this study, were 1) to determine the relationship between bilateral motor control brain activity and ACL injury risk biomechanics and 2) preliminarily isolate differences in bilateral motor control brain activity for those at high versus low risk of ACL injury.

## Materials and methods

This study was approved by the local Cincinnati Childrens Hospital Medical Center Institutional Review Board and informed consent was obtained from every participant. This investigation employed a leg press neuroimaging paradigm to examine brain activity for bilateral motor control. We adapted a unilateral version of this paradigm [33], specifically one with good to excellent intersession reliability over a ~seven week period [34], for bilateral lower extremity movement. Participant landing mechanics were also assessed during a drop vertical jump using traditional 3D motion analysis (separate from the neuroimaging paradigm). Knee abduction moments during landing were modelled as a covariate of interest within the fMRI analyses to identify directional relationships with leg press brain activity ('neural correlate' analysis below). A follow-up injury-risk classification analysis isolated differences in neural activity between high and low injury-risk subgroups, based on the established cut-off values for peak knee abduction moment during landing [3].

### Participants

This study enrolled pediatric female participants due to their greater relative ACL injury risk and increased propensity for reduced frontal plane knee control and lower extremity valgus alignment during landing and pivoting movements relative to males [35]. Thirty-one female high-school soccer players (Table 1) met inclusion criteria for neuroimaging and were

**Table 1. Group demographics and motor performance data.**

| Group | n | Age (years) | Height (cm) | Mass (kg) | BMI | Knee abduction moment (Nm) | Normalized Knee abduction moment (Nm\(kg*m) |
|---|---|---|---|---|---|---|---|
| Primary Knee Injury-Risk Correlate Analysis | | | | | | | |
| Full Cohort | 30 | 15.8±0.96 | 164.53 ± 5.76 | 58.14 ± 8.79 | 21.35 ± 2.78 | 15.0±10.10 | 0.16±0.10 |
| Secondary Injury-Risk Threshold Group Analysis | | | | | | | |
| High Injury-Risk (≥25.25 Nm peak knee abduction moment) | 4 | 16.2 ± 0.82 | 166.6 ± 4.83 | 63.7 ± 9.34 | 23.15 ± 3.46 | 32.88±6.10 | 0.31±0.04 |
| Low Injury-Risk (≤6.0 Nm peak knee abduction moment) | 5 | 16.4 ± 0.89 | 163.4 ± 5.90 | 60.08 ± 5.25 | 22.49 ± 1.36 | 2.80±2.54 | 0.03±0.027 |
| p-value | | 0.51 | 0.50 | 0.48 | 0.70 | < .001 | < .001 |

Demographics for each group in the injury-risk experiment, high injury-risk (≥25.25 Nm peak knee abduction moment), low injury-risk (≤10.6 Nm peak knee abduction moment). Knee abduction presented as absolute, and mass\height normalized.

evaluated using laboratory-based 3D motion analysis during a standardized drop vertical jump (DVJ) task. Of the thirty-one athletes, one was removed from analyses due to excessive head motion, and thirty were included in the neural correlate analysis. In the follow-up injury-risk classification analysis, sub groups of high and low injury risk were determined based on peak knee abduction moment; a highly reliable and commonly used metric for knee neuromuscular control and primary and secondary ACL injury risk [3, 30]. Four were classified into the high-risk classification of ≥25.25Nm peak knee abduction moment based on previous literature establishing this threshold as a prospective predictor of injury risk [3, 36]. Six were classified into the low-risk classification ≤6.00 Nm peak knee abduction moment. Five participants in the low-injury risk group were able to be matched to the high-injury risk group to minimize group-wise differences of age and body mass index (**Table 1**). Testing was completed over two days (visit 1, biomechanical landing assessment; visit 2: neuroimaging assessment) and all participants/parents signed written informed consent prior to completing MRI screening.

## Neuroimaging data acquisition & collection

Functional magnetic resonance imaging (fMRI) takes advantage of the blood oxygen level dependent (BOLD) signal as a surrogate to quantify neural activity [37]. Neuroimaging of the bilateral leg press task was performed using a 32-channel phased-array head coil. The MRI protocol included a 3-dimensional high-resolution T1-weighted image (repetition time: 8.3 ms, echo time: 3.7 ms, field of view: 256×256 mm; matrix: 256×256; slice thickness 1 mm, 176 slices) for image registration. fMRI data were acquired with a gradient-echo EPI sequence following a periodic block design in which the 30 s motor task (4 blocks) was interleaved with 30 s of rest (5 blocks) acquired with a 2 s repetition time, a 3.75×3.75 mm in-plane resolution, and a 5 mm slice thickness for 38 axial slices (field of view 240 mm and 64×64 matrix).

The bilateral leg press involved both a concentric press phase, an eccentric loading phase and a range of motion with greater ecological relevance to actual landing than prior methods, though still limited by neuroimaging constraints. The bilateral leg press task was from resting 0˚ full extension (without locking out the joint to avoid jerking movement) to approximately 45˚ knee flexion standardized at 0.6 Hz (**Fig 1**). This movement pace was found after extensive optimization experiments to minimize head motion. This motion was completed continuously for 30 seconds with 30 seconds of rest for 4 cycles. Each scan session started with 30 seconds of a blank screen. The subject saw a countdown of "2", "1", and then "MOVE" was displayed, then a metronome started to standardize the movement pace at 1.2 Hz. At the end of each movement block, the participant saw "2", "1", "STOP" to allow a gentle return to the rest position to minimize head motion during transitions. The movement was completed on a custom apparatus comprised of two separate foot pedals that run on tracks (**Fig 1**). The feet of the participant were strapped to the pedals and moved horizontally with flexion and extension of the

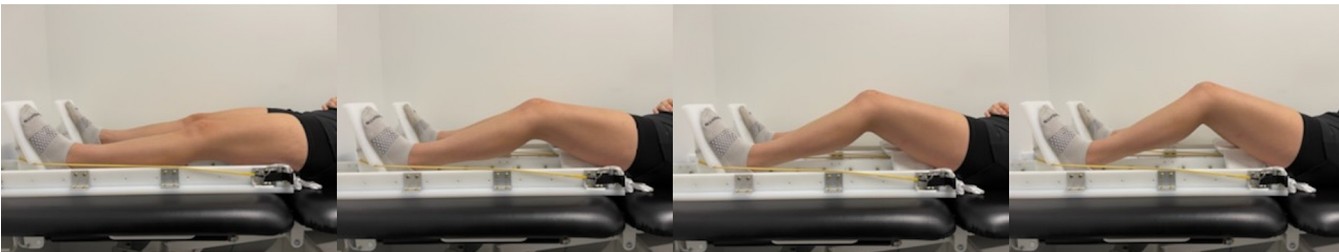

**Fig 1. Participant positioning to complete the bilateral leg press neuroimaging experiment.**

ankle, knee, and hip. An elastic resistance tube (manufacturer rated peak force ~9.1 kgs) was anchored at three points on the lateral side of both legs and in the center of the leg press apparatus providing independent tension for each leg throughout the movement. This resistance was found to be sufficient to stress bilateral lower extremity neuromuscular control without inducing additional head motion artifact. Through pilot testing, this resistance avoided excessive fatigue, due to repeated contractions in the block design, and minimized excessive head motion. Our goal was to utilize a bilateral leg press to partially resemble a landing movement pattern while considering and overcoming the limitations associated with fMRI/MRI scanning. This study was to our knowledge, the first attempt to stress bilateral lower extremity neuromuscular control similar to a drop landing by using simultaneous bilateral ankle, knee, and hip against resistance during brain fMRI.

A vital aspect of the study to ensure data quality, reliability, and minimal dropout due to head motion or participant discomfort was the completion of a mock scanner session before actual scanning, whereby the participants were trained on the bilateral leg press testing apparatus. First, a standardized video was played that explained and illustrated the task. Then, the participant had an opportunity to practice a full run of the task with guidance from the experimenter. All participants wore standardized shorts and socks without shoes to control for skin tactile feedback. Participants were positioned supine on the MRI table with customized padding and straps to reduce head motion. Handlebars were attached to the MRI table to standardize hand position and minimize accessory motion. This approach sufficiently reduced head motion during the motor paradigms with high reliability similar to our previously published unilateral lower extremity paradigm that exhibited good to high reliability for sensorimotor region activity [34].

## Drop vertical jump knee abduction moment

Three-dimensional kinematics and kinetics were recorded from participants performing the DVJ. During the DVJ, each participant fell forward off a 30 cm box, landed with both feet at the same time, and then immediately performed a vertical jump, raising both arms and reaching for a target set at 100% of their maximum jump height. Prior to the testing session, maximum counter movement vertical jump height was determined by having them attempt to grab a basketball at its maximum height on custom basketball retractor. Once ball height was raised to a height that ball could not be retrieved the highest successful measure was recorded and used to set the ball target height during the drop vertical jump testing. Participants completed practice trials to ensure task understanding and reduce learning effects. Then participants completed three separate trials of the DVJ and peak knee abduction moment during landing for each limb was calculated for each participant. The average of each of these six (three values for each knee) were averaged to represent participants' landing neuromuscular control.

Lower-limb joint kinematics were generated via the 3D trajectories of 42 externally mounted skin markers of 9 mm diameter attached to the athlete with double-sided adhesive tape. Marker trajectories were recorded via a 44-camera, high-speed (240 Hz) digital motion analysis system (Motion Analysis Corp.) and post-processed with Cortex software v. 6.0 (Motion Analysis Corp.). Prior to the performance of the DVJ, a static trial was recorded with all joints in neutral standing position. From the standing trial, a kinematic model comprised of 12 skeletal segments (upper arm ×2, lower arm ×2, trunk-thorax, pelvis, thigh ×2, shank ×2, & foot ×2) and 36 degrees of freedom was defined using Visual 3D software (v.6.01; C-Motion). Vertical ground reaction forces (VGRF) during landing were recorded via two embedded force platforms (AMTI, Inc.) sampled at 1200 Hz. Both VGRF and marker trajectories were low-pass filtered with a $4^{th}$-order cubic smoothing spline at a 12 Hz cut-off frequency. The

tracked 3D marker trajectories were processed through Visual3D to solve the generalized coordinates of the model for each frame. The VGRF data were used to normalize the kinematic data to 0–100% of stance, defined as the time period from initial contact with the force platforms to toe-off, with initial contact defined as the instant when VGRF first exceeds 10 N—at 1% increments (i.e., 101 data points). From the 3D kinematic and force plate data, 3D moments of the knee were computed using an inverse dynamics in Visual3D. Prior publications have established the strong validity and reliability of these 3D motion analysis methods for neuromuscular control and injury risk assessment [38]. An independent samples *t*-test was used to evaluate peak knee abduction moment differences between groups.

### Neuroimaging statistical analysis

Neuroimaging analyses were performed using the Oxford Centre for Functional MRI of the Brain software package, FSL 5.0.10 (FMRIB, Oxford UK) [39]. The data were spatially registered to correct for head motion artifact using MCFLIRT and spatially smoothed to improve sensitivity in quantifying functional activation as participants perform knee movements [39, 40]. This began with standard pre-processing applied to individual data, including non-brain removal, spatial smoothing using a Gaussian kernel of 5mm full width at half-maximum, and standard motion correction [41]. Realignment parameters (3 rotations and 3 translations) from the motion correction procedure were included in the design matrix as covariates to account for confounding effects of head movement. High-pass temporal filtering at 100 seconds and time-series statistical analysis were carried out using a linear model with local autocorrelation correction [42]. Functional images were co-registered with the respective high-resolution T1-weighted image and normalized to standard Montreal Neuroimaging Institute (MNI) 152 template using FNIRT non-linear image registration [41]. Thresholds for *z* scores were set at 3.1 and *p*-value at $p < .05$ cluster corrected for multiple comparison at the subject level (task contrast of move vs. rest) and group level analysis (group average for the task, correlate analysis and the sub-group injury risk group analysis). The neural correlate analyses were completed as a mixed-effects FLAME 1+2 model single group average one-sample t-test with knee abduction moment during landing demeaned and used as a covariate of interest to determine respective relationships to brain activity. The follow-up injury-risk classification analysis using high vs. low injury-risk participants was completed with a mixed-effects FLAME 1+2 model independent group contrast. Anatomical locations of significant clusters were identified using probabilistic maps derived from Harvard-Oxford and Juelich atlases, with regions exhibiting probabilities greater than 25% reported herein. As this was a brain activity correlate identification study, the effect size (r-value) of the relationship between brain activity and behavior are not reported to avoid circularity (voxel selection and magnitude estimation on the same data); a follow-up validation study is required to estimate effect size with the identified regions from this work [43, 44].

## Results

### Whole cohort task neural activity & association with knee abduction

Four brain clusters exhibited significant activation during the bilateral leg press paradigm relative to rest (**Table 2**, **Fig 2**). Greater knee abduction loading during the DVJ was associated with greater brain activation during the bilateral leg press in two clusters. Cluster 1 was located within the right lingual gyrus and intracalcarine cortex. Cluster 2 was in the bilateral lingual gyrus, posterior cingulate gyrus and precuneus (**Table 2**, **Fig 3**). No other significant relationships were observed. Participants' mean absolute and relative head motion was kept below 1

**Table 2. Task activity, injury-risk neural correlates and group classification analyses.**

| Cluster Index | Brain Regions | Voxel | P-value | Peak MNI Voxel | | | Z stat-max |
|---|---|---|---|---|---|---|---|
| | | | | x | y | z | |
| *Overall Activation During Bilateral Leg Press (n = 30)* | | | | | | | |
| 4 | Bilateral | 46591 | < .001 | 10 | -14 | 82 | 8.31 |
| | Precentral Gyrus | | | | | | |
| | Postcentral Gyrus | | | | | | |
| | Supplementary motor cortex | | | | | | |
| | Insular Cortex | | | | | | |
| | Opercular Cortex | | | | | | |
| | Planum Temporale | | | | | | |
| | Supramarginal Gyrus | | | | | | |
| | Cingulate | | | | | | |
| | Paracingulate | | | | | | |
| | Inferior, Middle, Superior Frontal Gyri, Pole | | | | | | |
| | Frontal Orbital Cortex | | | | | | |
| | Angular Gyrus | | | | | | |
| | Precuneus | | | | | | |
| | Inferior, Middle, Superior Temporal Gyri, Pole Temporal Fusiform | | | | | | |
| | Superior Parietal Lobule | | | | | | |
| | Lingual Gyrus | | | | | | |
| | Lateral Occipital Cortex | | | | | | |
| | Occipital pole | | | | | | |
| | Brainstem | | | | | | |
| | Putamen | | | | | | |
| | Caudate | | | | | | |
| | Cerebellum | | | | | | |
| 3 | Left | 362 | < .001 | -26 | -92 | -18 | 5.09 |
| | Occipital Pole | | | | | | |
| | Lateral Occipital Cortex | | | | | | |
| | Occipital Fusiform | | | | | | |
| | Lingual Gyrus | | | | | | |
| 2 | Right | 158 | .020 | 38 | -62 | -20 | 5.32 |
| | Temporal Occipital Fusiform | | | | | | |
| | Occipital Fusiform | | | | | | |
| | Inferior Temporal Gyrus | | | | | | |
| | Lateral Occipital Cortex | | | | | | |
| | Lingual gyrus | | | | | | |
| | Cerebellum | | | | | | |
| | Crus I | | | | | | |
| | VI | | | | | | |
| 1 | Right | 129 | .049 | 14 | 12 | -16 | 4.35 |
| | Frontal Orbital Cortex | | | | | | |
| | Parahippocampal Gyrus | | | | | | |
| | Subcallosal Cortex | | | | | | |
| *Neural Activity Positively Associated with Drop Vertical Jump Knee Abduction Moment (n = 30)* | | | | | | | |

(*Continued*)

**Table 2.** (Continued)

| Cluster Index | Brain Regions | Voxel | P-value | Peak MNI Voxel | | | Z stat-max |
|---|---|---|---|---|---|---|---|
| | | | | x | y | z | |
| 2 | Bilateral | 250 | < .001 | 0 | -54 | 16 | 4.84 |
| | Precuneus | | | | | | |
| | Lingual gyrus | | | | | | |
| | Intracalcarine cortex | | | | | | |
| | Posterior Cingulate | | | | | | |
| 1 | Right | 209 | .002 | 12 | -54 | 0 | 4.61 |
| | Lingual gyrus | | | | | | |
| | Intracalcarine cortex | | | | | | |
| | Precuneus | | | | | | |
| *Neural Activity High Injury-Risk Classification (n = 5) > Low Injury-Risk Classification (n = 5)* | | | | | | | |
| 3 | Bilateral | 1158 | < .001 | 4 | -62 | 10 | 5.54 |
| | Intracalcarine Cortex | | | | | | |
| | Precuneous | | | | | | |
| | Lingual gyrus | | | | | | |
| | Supracalcarine cortex | | | | | | |
| | Posterior Cingulate | | | | | | |
| 2 | Left | 181 | .002 | -14 | -48 | 0 | 4.64 |
| | Posterior Cingulate | | | | | | |
| | Lingual gyrus | | | | | | |
| 1 | Bilateral | 103 | .028 | 4 | -24 | 64 | 5.78 |
| | Supplementary Motor Cortex | | | | | | |
| | Precentral gyrus | | | | | | |

Areas of increased brain activity during the bilateral leg press relative to rest (top), activity associated with increased knee abduction moment during the drop vertical jump (middle) and injury-risk group classification (bottom). Voxel #: indicates number of activated voxels in this cluster. The clusters are identified statically using Gaussian random field theory to correct for multiple comparisons and identify the number of contiguous voxels whose voxel wise stats are above threshold MNI Montreal Neurologic Institute provides a standardized reference atlas for region location and identification. x, y, z indicates 3D location of voxel with highest activity level in the cluster. Z max: Z-statistic of the voxel with highest activity. Correlate region and sub-group threshold analysis are Harvard Oxford cortical, subcortical and cerebellar Atlas in MNI152 space FNIRT anatomical pre-threshold masked.

No regions were negatively associated with knee abduction moment or activated less in those with high injury risk neuromuscular control relative to low risk.

mm with average absolute head motion of 0.62 ± 0.30 and relative head motion was 0.14 ± 0.05.

## Neural activity differences between injury risk knee abduction loading group classification

In the sub-group injury-risk classification analysis, biomechanical analyses confirmed that the high-risk group had significantly higher absolute and normalized knee abduction moment (33.19±5.33 Nm) relative to the low-risk group (2.84 ± 1.58 Nm; $p < .001$) (**Table 1**). The high-risk group demonstrated significantly greater activation in three clusters relative to the low-risk group: cluster 1 encompassed the bilateral primary and premotor cortices, cluster 2 included the left posterior cingulate cortex and lingual gyrus, and cluster 3 was within the bilateral intracalcarine cortex, precuneus, and lingual gyrus (**Table 2**, **Fig 4**). There was no relative greater activity in the low injury-risk group compared to the high-risk group.

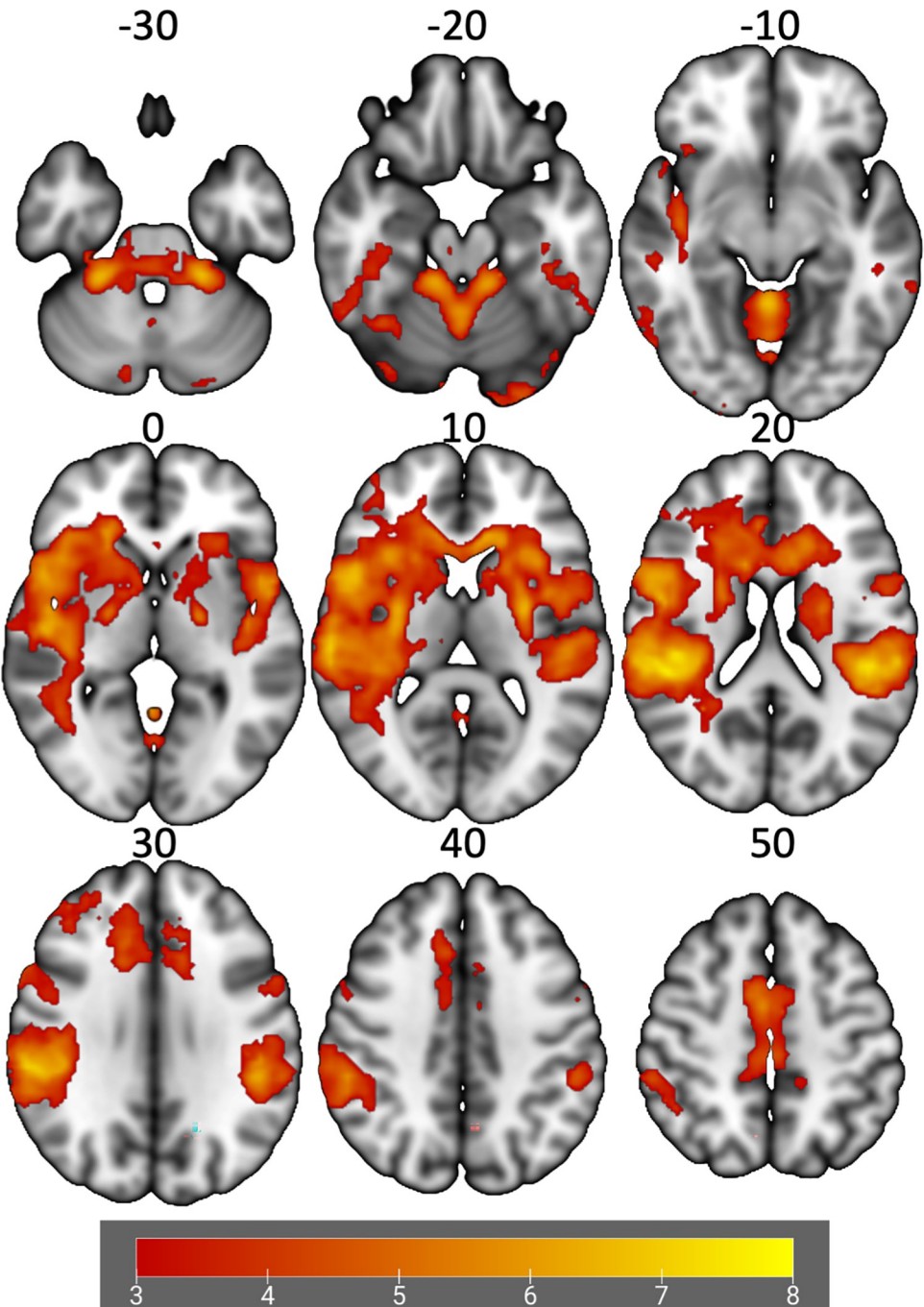

**Fig 2. Group average areas of increased activation during the bilateral leg press fMRI paradigm (z-statistic images with a cluster threshold of z > 3.1 and p < .05 corrected).** Numbers on top of each axial slice indicate the z-coordinate in MNI space and the bar on bottom reflects the z-statistic color map.

## Discussion

Drop landing peak knee abduction moment, a marker of ACL injury risk motor coordination in female athletes, was positively associated with bilateral motor control brain activity in sensory integration regions. Specifically, greater knee abduction loading was associated with

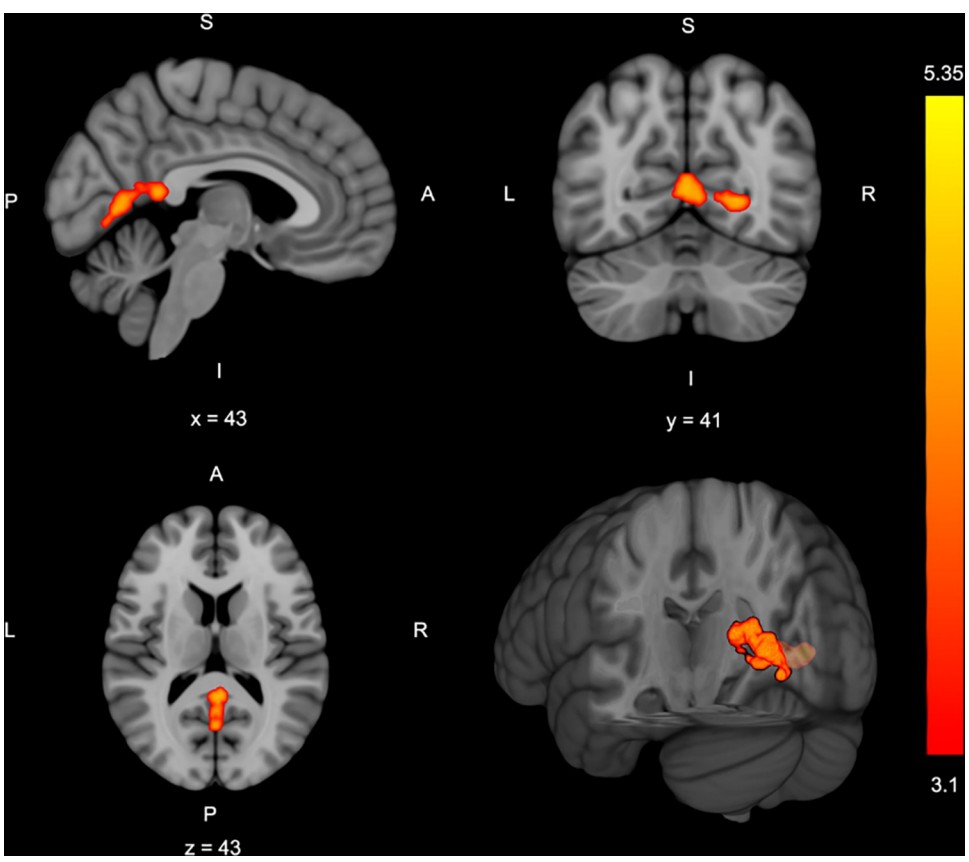

**Fig 3. Group neural correlate z-statistic map (cluster threshold of z > 3.1, p < .05, corrected), depicting regions where greater brain activity was associated with greater knee abduction moment during drop landing.** Slice location indicated by x, y or z location and S: Superior; L: Left; R: Right; I: Inferior; A: Anterior; P: Posterior for image orientation. The bar to right indicates the z-statistic color map.

greater $p_{recuneus}$ (area integrating sensorimotor coordination when visual-spatial attention is required) [45, 46], posterior cingulate cortex (area processing spatial awareness and attention for motor control) [45, 47, 48], and lingual gyrus and intracalcarine cortex activity (areas processing spatial awareness and attention for motor control) [49–51] activity (**Table 2**, **Fig 3**). Further, those classified at high risk for injury had similar regions of greater activity with the addition of more widespread primary and secondary sensorimotor cortex activity when compared to those classified as low injury-risk (**Table 2**, **Fig 4**).

### A neural correlate for injury risk—Visuospatial and sensory integration neural activity

The relationship between landing biomechanics and bilateral motor control brain activity may indicate that injury-risk biomechanics are, in part, a manifestation of reduced neural crossmodal sensory integration activity for knee spatial awareness, potentially contributing to an inability to control knee abduction motion and loading. This assertion is supported not only by the crossmodal processing of the lingual gyrus [51–54] but also by the implicated sensory integration brain regions that provide the motor cortex with vital sensory, visuospatial, and attentional information to prepare and fine-tune motor action [45–47]. The greater sensory and visuospatial information processing requirements to engage in bilateral lower extremity motor control may reduce the ability to maintain a safe knee position during more dynamic

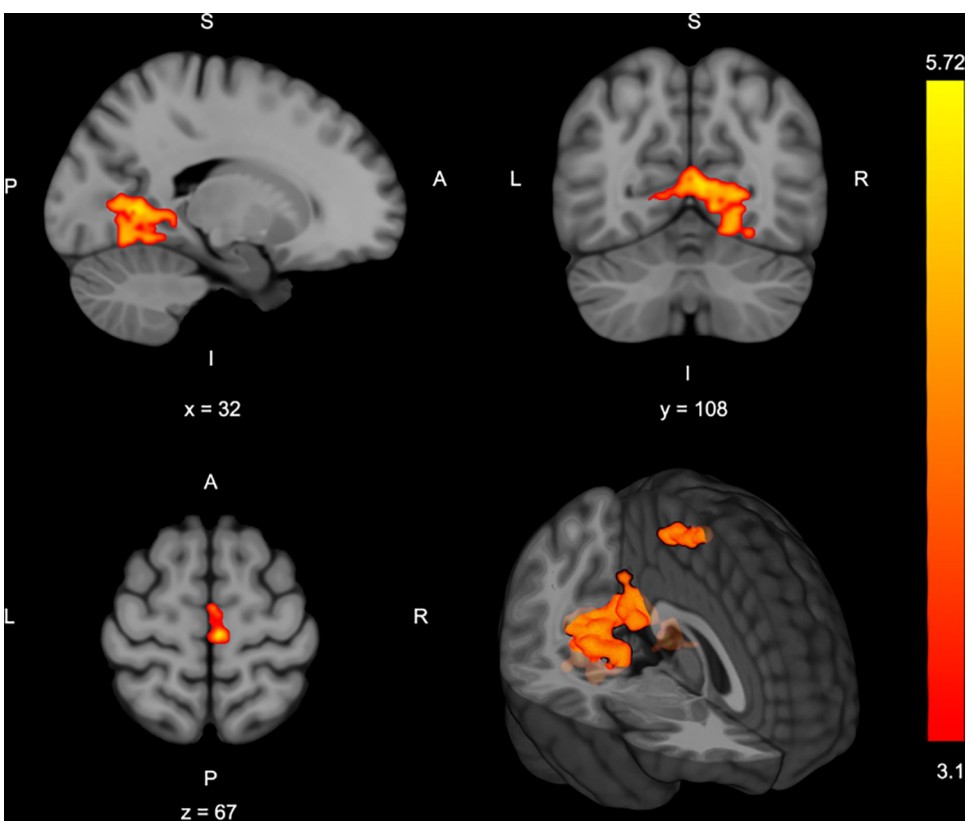

**Fig 4. Areas of greater brain activity in those classified as high risk based on drop landing knee abduction moment (z-statistic images with a cluster threshold of z > 3.1 and p < .05 corrected).** The bar to right indicates z-statistic for the activity difference between groups Slice location indicated by x, y or z location and S:superior; L: left; R: right; I: inferior; A: anterior; P: posterior for image orientation.

maneuvers while navigating a complex environment. Specifically, athletes with high injury-risk landing biomechanics may not have available neural resources to both maintain neuro-muscular integrity and engage in activities that require visuospatial and/or sensory processing (e.g., opponents, balls, goal; when ACL injuries often occur [14, 15]. Although the bilateral leg press task did not include such external variables to stress visuospatial and/or sensory processing, greater activation under a controlled bilateral movement may translate to reduced neural capacity or inefficiency when under instances of greater cognitive or sensory complexity such as the sporting environment.

Inefficient brain activation during functional tasks, as demonstrated by those with higher injury-risk movement coordination, may be the critical CNS link between visuospatial and cognitive attention deficits that increase ACL injury-risk [55, 56]. While we did not measure visuospatial and/or cognitive abilities, a connection between decreased visual reaction time and cognitive processing ability with greater knee abduction loading and motion has been established [57, 58]. Thus, it is possible that increased neural demand in visuospatial navigation and sensory integration brain regions for knee motor control may underly or mediate the relationship between visual attention abilities and injury risk movement coordination. Specifically, the identified lingual gyrus brain activity suggests crossmodal processing (congruent integration of multiple sensory modalities, proprioceptive and visuospatial in this case) [51, 52] is elevated in those with reduced motor coordination ability and injury-risk loading and could be a neural activity link between visual cognitive abilities and ACL injury risk.

The present data also share overlap with a recent experiment investigating the relationship of brain activity during a unilateral leg press task, as assessed with fMRI, and frontal plane range of motion (operationalized as poor frontal plane motor control) quantified concurrently with MR-safe 3D motion analysis [33]. Greater brain activity in regions important for cognition, sensorimotor control, and sensorimotor integration were associated with greater real-time frontal plane range of motion. Interestingly, there was partial anatomical overlap and shared directionality between the present findings and this prior report, specifically poorer knee frontal plane control (greater knee abduction moment during DVJ and greater frontal range of motion during the leg press) was associated with greater posterior cingulate and precuneus brain activity. However, the brain activity was not overlapping for all region associations as the present experiment employed a bilateral task with unique neural demands relative to the prior unilateral paradigm [59]. Further, we hypothesize brain activity correlates of biomechanics during landing in a traditional laboratory setting to be a better representation of the brain activity underlying ACL injury risk neuromuscular control and a more viable neural targets for screening and prevention methods. While the concurrent kinematics of frontal plane ROM during the unilateral leg press may be more indicative of subtle motor control variation. However, the overlapping sensory integration neural activity corresponding to increased injury-risk and reduced frontal plane knee control do indicate further examination of sensory, spatial and crossmodal processing as areas for further research.

## Injury risk classification associated with visuospatial and sensorimotor neural activity

Complementing the neural correlate analyses, the injury-risk threshold sub-group analysis uniquely identified sensorimotor control brain regions, precisely the premotor cortex and primary motor cortex, had greater activation in those with high-risk landing mechanics relative to those with low injury-risk landing mechanics. Specific to motor tasks, increased primary and secondary motor cortex activity can be caused by increased neuronal firing rates required to generate higher muscle forces, movement velocity, or manage movement complexity [25]. Additionally, as relative movement complexity increases (e.g., increased movement speed, force or attentional demand), brain activity in primary and pre-motor cortices also increases to sustain performance [25, 60]. While we did not modify task complexity, greater brain activity in those with higher injury-risk landing neuromuscular control may represent relative increases in neurological complexity and demand to engage in bilateral coordination of the lower extremity. This may indicate those with high knee abduction loading during the DVJ may be disproportionately taxed at a higher motor complexity threshold, thus more susceptible to coordination errors, excessive knee valgus, and subsequent injury when going from the isolated DVJ to athletic activity [55, 61].

This hypothesis of relative task complexity is further supported by the high injury-risk group brain activation pattern being similar to "novice" motor performers who depend on increased brain activity for motor planning and engage in an overly visuocognitive dominant strategy [62–65]. Thus, those with high-risk movement patterns may be "novice" in terms of the motor skill of maintaining knee alignment. Alternatively, the low-risk group appeared to adopt an "expert" motor activation strategy relative to the high injury-risk group by not increasing brain activity (indicating potential neural efficiency) to engage in the bilateral leg press task and control knee position and load [66]. A similar response has been reported in highly trained athletes when executing motor tasks [66]. In these studies, high-level karate athletes required less brain activation to stand on one leg compared to non-athletic counterparts, suggesting extensive training for single-leg stability makes the task neurologically less demanding over time. Applied to our data, those with low-risk neuromuscular control have a similar

"expert" or more efficient knee control strategy allowing additional cognitive or motor demands to be attenuated during sport and avoid high injury-risk positions [67].

Interestingly, a similar increase in brain activity within motor and visuospatial regions has been shown in those with a history of ACL injury relative to controls [19, 20]. In one of the few neural correlate studies of a biomechanical measure after ACLR, Chaput et al. indicated a potential sensory integration neural pathway (via posterior cingulate and precuneus activity) to enable visual cognitive ability to compensate and maintain proprioceptive acuity and jump landing stability after ACLR [68]. As the posterior cingulate and precuneus were also identified to contribute to injury-risk landings herein, it is possible that differences in brain activity previously attributed to injury, may in part, contribute to the initial injury event as well.

## Implications for injury reduction efforts

The brain activation profile for those with high injury-risk mechanics indicates distinct neural mechanisms involved in sensorimotor, visual-proprioceptive and spatial processing that could be used to supplement contemporary strategies for ACL injury reduction programs. For instance, applying motivation-based motor learning strategies may be uniquely capable of restoring athletes' CNS functioning for injury-resistant motor control through neurophysiologic-based mechanisms [69–71]. Specifically, providing athletes with a sense of autonomy and safely enhancing their confidence in a prescribed task may promote dopaminergic transmission via motivational mechanisms, for robust, adaptive neuroplasticity [72]. Further, directing an athlete's attention towards the effects of their movement on the environment (i.e., an external focus; opposed to an internal focus on the athlete's movements) has been shown to enhance motor behavior [72], including biomechanics associated with ACL injury risk [73–75]. Adopting an external focus strategy modulates inhibitory circuits within the primary motor cortex [76, 77] and can alter knee sensorimotor brain activity [78]. As such, an external focus of attention employed over the course of an ACL injury reduction program may be particularly well suited to better target the sensorimotor brain activity found in those with high-risk biomechanics [79]. Indeed, augmented reality visual biofeedback systems that utilize principles of motor learning have shown preliminary efficacy in targeting brain activity identified to be associated with high-injury risk and enhance injury resistant motor control when integrated into established ACL injury prevention methods [80, 81]. Specifically, improvements in athlete biomechanics following six weeks of augmented neuromuscular training was related to more efficient sensorimotor-related brain activity for unilateral knee motor control [81] and strengthened functional connectivity between knee sensorimotor control brain regions at rest [80].

Those with high injury-risk coordination also exhibit a brain activation pattern shifted toward visual-proprioceptive and spatial processing to organize movement, thus strategies that reduce reliance on vision for motor control may be useful to enhance ACL injury reduction efforts. For example, stroboscopic glasses are a novel ancillary tool that allow clinicians to incrementally increase the level of visual perturbation [82, 83]. Whereas clinicians were previously limited to eyes-open and eyes-closed conditions, stroboscopic glasses allow clinicians to perturb an athlete's vision during any exercise to varying degrees [84]. This type of visual perturbation can increase the proprioceptive integration demand during motor control, potentially similar to the dynamic sport environment [85, 86]. Training with these glasses have demonstrated increased visual processing efficiency that may improve the athlete's ability to navigate the environment [87]. Collectively, the high injury-risk brain activation pattern, while preliminary, may support novel approaches to enhance ACL injury risk reduction programs that enhance intervention efficacy and reduce the high dose required for effectiveness.

## Limitations

The current investigation utilized the literature-derived metric of knee abduction moment as the biomechanical variable related to injury risk due to its prospective association with injury in female athletes and similarity to the injury mechanism. However, we acknowledge other biomechanical risks factors have been identified for ACL injury risk, including asymmetrical ground reaction forces, excessive hip internal rotation and decreased knee and hip flexion [3, 36]. Therefore, brain activity specifically associated with knee abduction moment may not be sufficient to fully encompass the brain activity associated with aberrant neuromuscular control in females more broadly. Future investigations are warranted to investigate brain activity associated with a more comprehensive biomechanical profile of neuromuscular control. Also, the results of this study isolated neural signatures associated with known female sex-specific biomechanical risk factors for ACL injury. Thus, the present findings may not, or only partially, generalize to males. We would hypothesize that altered sensorimotor-related brain activity would be related to the known male sex-specific biomechanics associated with injury risk, such as decreased trunk flexion [88] and landing ground reaction asymmetry [89]. The fMRI leg press is also a simplified assay of motor control and more sophisticated force, or position matching paradigms may increase sensitivity to detect neural activity associated with movement regulation. It is important to note that we used the DVJ as an assay of neuromuscular control to examine relative brain activity and to dichotomize groups, not to directly implicate the fMRI paradigm as a replacement for the DVJ. Additionally, our fMRI task of a bilateral leg press is not representative of all the factors involved in a landing task (participant must lay supine with no head motion during active scanning) and participant biomechanics were not quantified concurrently with the neuroimaging paradigm. Integrating MRI-compatible 3D motion analysis systems [33, 90] could supplement the current findings by isolating whether those with increased frontal plane loading during the DVJ also present with lesser frontal plane control during the bilateral leg press. However, we consider the present, multi-joint bilateral leg press against resistance an advancement relative to previous fMRI paradigms. As this was a preliminary investigation on a novel fMRI motor task and brain-behavioral correlate, the small sample size is a limitation, though is in line with prior fMRI sample sizes for similar tasks [24, 25, 91], and will allow for effect size calculation for future larger studies. Future research can build on the current results by enhancing the current fMRI paradigm to better simulate the cognitive-perceptual demands of dynamic sporting environments while adjusting the task constraints to simulate more realistic landing mechanics within fMRI. Specifically, follow-up studies could incorporate MRI-compatible virtual reality to provide external visual stimuli while moving (e.g., a virtual ball or defender) and increase the current paradigms metronome speed with increased dynamic lower extremity loading to better replicate the biomechanical demands of an actual landing.

## Conclusion

Distinct brain activity in regions that integrate visual, proprioceptive, and attentional information may underlie ACL injury-risk landing biomechanics in pediatric females. While confirmatory studies are warranted, the present data indicate potential neural correlates for female athlete ACL injury risk neuromuscular control, and a novel pathway for the optimization of ACL injury risk reduction strategies.

## Supporting information

**S1 Data.**
(XLSX)

## Author Contributions

**Conceptualization:** Dustin R. Grooms, Jed A. Diekfuss, Cody R. Criss, Manish Anand, Gregory D. Myer.

**Data curation:** Dustin R. Grooms, Jed A. Diekfuss, Manish Anand, Alexis B. Slutsky-Ganesh, Christopher A. DiCesare, Gregory D. Myer.

**Formal analysis:** Dustin R. Grooms, Jed A. Diekfuss, Cody R. Criss, Manish Anand, Alexis B. Slutsky-Ganesh, Christopher A. DiCesare.

**Funding acquisition:** Dustin R. Grooms, Jed A. Diekfuss, Gregory D. Myer.

**Investigation:** Dustin R. Grooms, Jed A. Diekfuss, Cody R. Criss, Manish Anand, Alexis B. Slutsky-Ganesh, Christopher A. DiCesare, Gregory D. Myer.

**Methodology:** Dustin R. Grooms, Jed A. Diekfuss, Manish Anand, Alexis B. Slutsky-Ganesh, Christopher A. DiCesare, Gregory D. Myer.

**Project administration:** Dustin R. Grooms, Jed A. Diekfuss, Christopher A. DiCesare, Gregory D. Myer.

**Resources:** Jed A. Diekfuss, Gregory D. Myer.

**Software:** Manish Anand.

**Supervision:** Dustin R. Grooms, Jed A. Diekfuss, Gregory D. Myer.

**Validation:** Dustin R. Grooms, Cody R. Criss, Manish Anand, Alexis B. Slutsky-Ganesh.

**Visualization:** Dustin R. Grooms, Cody R. Criss, Manish Anand, Alexis B. Slutsky-Ganesh.

**Writing – original draft:** Dustin R. Grooms.

**Writing – review & editing:** Dustin R. Grooms, Jed A. Diekfuss, Cody R. Criss, Manish Anand, Alexis B. Slutsky-Ganesh, Christopher A. DiCesare, Gregory D. Myer.

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
