## [Decision Letter · Decision Letter 0]

11 Feb 2022

PONE-D-21-23995Preliminary neural correlates of anterior cruciate ligament injury-risk biomechanicsPLOS ONE

Dear Dr. Grooms,

Thank you for submitting your manuscript to PLOS ONE. After careful consideration, we feel that it has merit but does not fully meet PLOS ONE’s publication criteria as it currently stands. Therefore, we invite you to submit a revised version of the manuscript that addresses the points raised during the review process.

We look forward to receiving your revised manuscript.

Kind regards,

Riccardo Di Giminiani

Academic Editor

PLOS ONE

Journal Requirements:

5. We note that Figure 1 in your submission contain copyrighted images. All PLOS content is published under the Creative Commons Attribution License (CC BY 4.0), which means that the manuscript, images, and Supporting Information files will be freely available online, and any third party is permitted to access, download, copy, distribute, and use these materials in any way, even commercially, with proper attribution. For more information, see our copyright guidelines: http://journals.plos.org/plosone/s/licenses-and-copyright.

Reviewers' comments:

Reviewer's Responses to Questions

**Comments to the Author**

1. Is the manuscript technically sound, and do the data support the conclusions?

Reviewer #1: Partly

Reviewer #2: Yes

2. Has the statistical analysis been performed appropriately and rigorously? 

Reviewer #1: Yes

Reviewer #2: Yes

3. Have the authors made all data underlying the findings in their manuscript fully available?

Reviewer #1: Yes

Reviewer #2: Yes

4. Is the manuscript presented in an intelligible fashion and written in standard English?

Reviewer #1: Yes

Reviewer #2: Yes

5. Review Comments to the Author

Reviewer #1: General comments

The aim of this study was to study the association between bilateral motor control brain activity (assessed via fMRI, during a leg press dynamic task) and a specific ACL injury risk biomechanical parameter, i.e. knee abduction moment during landing from a drop vertical jump, in a population of pediatric females (n = 30).

The main findings of the study are the following: 1) greater landing knee abduction moments were associated with greater brain activity in specific regions that integrate visual, proprioceptive and attentional information; 2) the subgroup of participants classified as at higher ACL injury risk had greater brain activity with respect to those at lower injury risk.

The study is original and has been conducted with methodological rigor.

I offer suggestions for methodological clarifications, data reporting and editorial amendments.

Specific comments

LL 29. If possible I would suggest to add details regarding the leg-press load (e.g. %)

LL 106. The choice of studying pediatric female participants because of their “greater relative ACL injury risk, increased propensity for reduced frontal plane knee control, lower extremity valgus alignment during landing and pivoting moments” relative to males has been properly justified in the methods. However, the emphasis of the entire introduction (as well as in the abstract, discussion and conclusion) seem not to be specific to the studied population, but rather it focuses on the fact that ACL rupture is “a common sport-or physical activity-related injury among adolescent and young athletes”. My question is the following: has this analysis been made on female participants only? And more importantly, could the same results be expected for males also? This point should be probably addressed in the discussion section, where results seem to be discussed and generalized to the whole population level, rather than to be specific to the involved population.

LL 119. Please be consistent when using terminology: high-risk participants (in the text) vs. high-injury risk (Table 1). Same for low-risk group (text) vs. low-injury risk (Table 1).

LL 120. It is stated that the “pairwise matching” between low-risk group and high-risk group participants was based on age and on body mass index. However, BMI data are not reported in table 1. Moreover, from table 1 it seems that the high-risk group had similar age, but was slightly taller and much heavier than the low-injury risk group. Therefore, how was the matching performed? Furthermore, how were the 5 participants out of 11 selected/identified for the pairwise matching analysis. Please clarify.

LL 128. Sometimes it is used “body mass” (L 151), sometimes only “mass” (L 128) or “weight” (Table1). Please be consistent throughout the text.

LL 138. “The bilateral leg press involved an eccentric loading phase and…”. From this sentence, it seems that fMRI was recorded only during the eccentric phase, whereas from LL 142-146 it is clear that participants performed both contraction phases (eccentric + concentric) continuously for 30 s (from full extension to 45° knee flexion). Consider rephrasing.

LL 138 and 171. Were participants familiarized with both bilateral leg press exercise and with drop vertical jump?

LL 151. How was the elastic resistance standardized? Why was 35% of body mass selected? Despite the necessity to keep a low resistance to avoid head artifacts, the 35% of participant’s body mass seems to be quite a low resistance to assess, also if we consider the potential forces acting on the knee during a drop vertical jump.

LL 181-191. This entire section provides detailed indication of parameters extracted from the drop vertical Jump (e.g. VGRF). However, these variables related to DVJ data have not been described in the results.

LL 221. It is not clear where the “RESULTS” section starts, as this is not indicated in the manuscript. As it is, it seems that Methods and Results are described in the same section on a continuum.

LL 230. Please be more specific when describing the figure caption. What do the numbers on top of each image indicate? What is the unit of measure of the yellow-orange bar at the bottom? I assume that all the figures are representative of one participant, presumably with an high ACL injury risk.

LL 231. Refer to comment LL 230.

LL 238-239. It seems that that in the sentences starting with “Cluster 1” and “Cluster 2” the main verbal tense is missing.

LL 247. Please be consistent when reporting the number of decimals in the p-value (sometimes you report 5 decimals, sometimes 4, sometimes 2, sometimes 3).

LL 267. Is “drop landing peak knee abduction moment” a specific marker for young female athletes? If so, it would be better to specify it. In this way, also your choice of studying females only could be better justified.

LL 303. The reference paper to this “recent experiment” is only reported 10 rows below. I would suggest to add this reference either here or to move it up a bit in the text for better clarity.

LL 387. The fact that the association was studied in pediatric females only should be addressed in the limitations.

LL 415-417. The fact that the association between brain activity and this specific ACL injury risk biomechanics parameter has been studied and observed in pediatric females only, may not support and justify such a broad and generalized conclusion.

Reviewer #2: Overall, the study is very well structured, detailed in the methodology, in line with the most recent evidences and provides important information to deepen the knowledge about the underlying causes that increase the risk of sustaining an ACL injury. This was achieved by correlating known biomechanical risk factors (knee abduction moment) with brain activity in different areas, analyzing for the first time a two-legged task such as the bilateral leg press. Personally, I believe that the study is highly suitable for publication.

Below some minor remarks:

Methods

Page 5 Line 108: How did you select the sample size? Did you perform an apriori sample size calculation? Please argue on that

Page 6 Lines 139-140: “thereby facilitating a greater correspondence with actual neuromuscular control demands of the DVJ”. I suggest removing this sentence since the speculation may send out an ambiguous message considering the complexity and the real neuromuscular-coordinative demand of a jump-landing maneuver with respect to the bilateral leg press task.

Page 8 Line 174: How did you measure maximum jump height? Please argue on that

Discussion

Page 13 line 280: Please change “This assertion is support not only by” with “This assertion is supported not only by”

Page 13 line 281: Please change “but the implicated sensory” with “but also by the implicated sensory”

Limitations

Please add the small sample size as a limitation to the generalizability of the study results.

6. PLOS authors have the option to publish the peer review history of their article (what does this mean?). If published, this will include your full peer review and any attached files.

Reviewer #1: No

Reviewer #2: **Yes: **jacopo emanuele rocchi

---

## [Author Response · Author response to Decision Letter 0]

14 Apr 2022

This is also included in the uploaded file

Review Comments to the Author

Reviewer #1: General comments

The aim of this study was to study the association between bilateral motor control brain activity (assessed via fMRI, during a leg press dynamic task) and a specific ACL injury risk biomechanical parameter, i.e. knee abduction moment during landing from a drop vertical jump, in a population of pediatric females (n = 30).

The main findings of the study are the following: 1) greater landing knee abduction moments were associated with greater brain activity in specific regions that integrate visual, proprioceptive and attentional information; 2) the subgroup of participants classified as at higher ACL injury risk had greater brain activity with respect to those at lower injury risk.

The study is original and has been conducted with methodological rigor. I offer suggestions for methodological clarifications, data reporting and editorial amendments.

RESPONSE: Thank you for your critical review and suggestions to improve the manuscript. 

Specific comments

LL 29. If possible I would suggest to add details regarding the leg-press load (e.g. %)

RESPONSE: We have revised the abstract and provide further explanation on the resistance in the manuscript:

“The feet of the participant were strapped to the pedals and moved horizontally with flexion and extension of the ankle, knee, and hip. An elastic resistance tube (manufacturer rated peak force ~9.1 kgs) was anchored at three points on the lateral side of both legs and in the center of the leg press apparatus providing independent tension for each leg throughout the movement. This resistance was found to be sufficient to stress bilateral lower extremity neuromuscular control without inducing additional head motion artifact. Through pilot testing, this resistance avoided excessive fatigue, due to repeated contractions in the block design, and minimized excessive head motion. Our goal was to utilize a bilateral leg press to partially resemble a landing movement pattern while considering and overcoming the limitations associated with fMRI/MRI scanning. This study was to our knowledge, the first attempt to stress bilateral lower extremity neuromuscular control similar to a drop landing by using simultaneous bilateral ankle, knee, and hip against resistance during brain fMRI.”

LL 106. The choice of studying pediatric female participants because of their “greater relative ACL injury risk, increased propensity for reduced frontal plane knee control, lower extremity valgus alignment during landing and pivoting moments” relative to males has been properly justified in the methods. However, the emphasis of the entire introduction (as well as in the abstract, discussion and conclusion) seem not to be specific to the studied population, but rather it focuses on the fact that ACL rupture is “a common sport-or physical activity-related injury among adolescent and young athletes”. My question is the following: has this analysis been made on female participants only? And more importantly, could the same results be expected for males also? This point should be probably addressed in the discussion section, where results seem to be discussed and generalized to the whole population level, rather than to be specific to the involved population.

RESPONSE: This is an excellent point, and we agree, the discussion required recognition of the specific implications of this work to the female athlete. We have added female specific clarifiers where appropriate and added this to the limitations regarding males: 

“Also, the results of this study isolated neural signatures associated with known female sex-specific biomechanical risk factors for ACL injury. Thus, the present findings may not, or only partially, generalize to males. We would hypothesize that altered sensorimotor-related brain activity would be related to the known male sex-specific biomechanics associated with injury risk, such as decreased trunk flexion[88] and landing ground reaction asymmetry[89].”

 LL 119. Please be consistent when using terminology: high-risk participants (in the text) vs. high-injury risk (Table 1). Same for low-risk group (text) vs. low-injury risk (Table 1).

RESPONSE: Thank you for pointing this out, the discrepancy is now corrected. 

LL 120. It is stated that the “pairwise matching” between low-risk group and high-risk group participants was based on age and on body mass index. However, BMI data are not reported in table 1. Moreover, from table 1 it seems that the high-risk group had similar age, but was slightly taller and much heavier than the low-injury risk group. Therefore, how was the matching performed? Furthermore, how were the 5 participants out of 11 selected/identified for the pairwise matching analysis. Please clarify.

RESPONSE: Thank you for pointing this out, we have revised the text to be clearer on the group-wise inclusion, matching and independent t-test group analysis based on injury-risk classification. We now include BMI in the demographics table and report the t-test for knee abduction moment as an absolute variable and normalized to height and mass.

LL 128. Sometimes it is used “body mass” (L 151), sometimes only “mass” (L 128) or “weight” (Table1). Please be consistent throughout the text.

RESPONSE: Thank you for catching this, we now only report as “mass” throughout. 

LL 138. “The bilateral leg press involved an eccentric loading phase and…”. From this sentence, it seems that fMRI was recorded only during the eccentric phase, whereas from LL 142-146 it is clear that participants performed both contraction phases (eccentric + concentric) continuously for 30 s (from full extension to 45° knee flexion). Consider rephrasing.

RESPONSE: Thank you for the opportunity to clarify – you are correct the fMRI was recorded during the entire movement cycle – revised as requested and the new text is included below:

“The bilateral leg press involved both a concentric press phase, an eccentric loading phase and a range of motion with greater ecological relevance to actual landing than prior methods, though still limited by neuroimaging constraints.”

LL 138 and 171. Were participants familiarized with both bilateral leg press exercise and with drop vertical jump?

RESPONSE: Yes, a familiarization session was completed for the fMRI leg press as now further clarified and practice trials were given for the drop vertical jump as now described.

“First, a standardized video was played that explained and illustrated the task. Then, the participant had an opportunity to practice a full run of the task with guidance from the experimenter”

“Participants completed practice trials to ensure task understanding and reduce learning effects.”

LL 151. How was the elastic resistance standardized? Why was 35% of body mass selected? Despite the necessity to keep a low resistance to avoid head artifacts, the 35% of participant’s body mass seems to be quite a low resistance to assess, also if we consider the potential forces acting on the knee during a drop vertical jump.

RESPONSE: This is a fair point and a limitation of modern neuroimaging technology. We have clarified the band resistance in the text: 

“The feet of the participant were strapped to the pedals and moved horizontally with flexion and extension of the ankle, knee, and hip. An elastic resistance tube (manufacturer rated peak force ~9.1 kgs) was anchored at three points on the lateral side of both legs and in the center of the leg press apparatus providing independent tension for each leg throughout the movement. This resistance was found to be sufficient to stress bilateral lower extremity neuromuscular control without inducing additional head motion artifact. Through pilot testing, this resistance avoided excessive fatigue, due to repeated contractions in the block design, and minimized excessive head motion. Our goal was to utilize a bilateral leg press to partially resemble a landing movement pattern while considering and overcoming the limitations associated with fMRI/MRI scanning. This study was to our knowledge, the first attempt to stress bilateral lower extremity neuromuscular control similar to a drop landing by using simultaneous bilateral ankle, knee, and hip against resistance during brain fMRI"

LL 181-191. This entire section provides detailed indication of parameters extracted from the drop vertical Jump (e.g. VGRF). However, these variables related to DVJ data have not been described in the results.

RESPONSE: We have revised this section and removed unneeded methods details. We also clarify the calculation of knee abduction moment through inverse dynamics uses ground reaction force data as defined in prior literature.

LL 221. It is not clear where the “RESULTS” section starts, as this is not indicated in the manuscript. As it is, it seems that Methods and Results are described in the same section on a continuum.

RESPONSE: Thank you for pointing this out, we now include a header for the results section and break up the results by the whole group neural correlate analysis and the sub-group difference analysis. 

LL 230. Please be more specific when describing the figure caption. What do the numbers on top of each image indicate? What is the unit of measure of the yellow-orange bar at the bottom? I assume that all the figures are representative of one participant, presumably with a high ACL injury risk.

RESPONSE: The figure captions have been revised to fully describe the image and are group averages (not individual participant data and this is now clarified).

LL 231. Refer to comment LL 230.

RESPONSE: The figure captions have been revised to fully describe the image. 

LL 238-239. It seems that that in the sentences starting with “Cluster 1” and “Cluster 2” the main verbal tense is missing.

RESPONSE: We have revised the structure of the sentence.

LL 247. Please be consistent when reporting the number of decimals in the p-value (sometimes you report 5 decimals, sometimes 4, sometimes 2, sometimes 3).

RESPONSE: Thank you for catching this, the p-values are now revised to three decimal places throughout the manuscript.

LL 267. Is “drop landing peak knee abduction moment” a specific marker for young female athletes? If so, it would be better to specify it. In this way, also your choice of studying females only could be better justified.

RESPONSE: Yes, that is correct knee abduction moment (or the joint moments that push the knees to collapse toward midline during landing, resulting in a dynamic valgus alignment) is a specific motor coordination marker for the female athlete. We have revised the opening discussion sentence to reinforce this point before summarizing the neural results. 

LL 303. The reference paper to this “recent experiment” is only reported 10 rows below. I would suggest to add this reference either here or to move it up a bit in the text for better clarity.

RESPONSE: We have added the reference here for clarity. 

LL 387. The fact that the association was studied in pediatric females only should be addressed in the limitations.

RESPONSE: We agree and have clarified the data is specific to female athletes and included the new text below:

“Also, the results of this study isolated neural signatures associated with known female sex-specific biomechanical risk factors for ACL injury. Thus, the present findings may not, or only partially, generalize to males. We would hypothesize that altered sensorimotor-related brain activity would be related to the known male sex-specific biomechanics associated with injury risk, such as decreased trunk flexion[88] and landing ground reaction asymmetry[89].”

LL 415-417. The fact that the association between brain activity and this specific ACL injury risk biomechanics parameter has been studied and observed in pediatric females only, may not support and justify such a broad and generalized conclusion.

RESPONSE: We agree and have added pediatric female athlete into the conclusion to reduce over generalization. 

Reviewer #2: Overall, the study is very well structured, detailed in the methodology, in line with the most recent evidences and provides important information to deepen the knowledge about the underlying causes that increase the risk of sustaining an ACL injury. This was achieved by correlating known biomechanical risk factors (knee abduction moment) with brain activity in different areas, analyzing for the first time a two-legged task such as the bilateral leg press. Personally, I believe that the study is highly suitable for publication.

RESPONSE: Thank you for this comment and your constructive critiques to improve the manuscript. 

Below some minor remarks:

Methods

Page 5 Line 108: How did you select the sample size? Did you perform an apriori sample size calculation? Please argue on that

RESPONSE: We did not employ an apriori sample size calculation as this preliminary study was undertaken to support the development of effect sizes for our outcomes of interest. As this is a novel task in the fMRI literature and a novel scientific question, we did not have direct reports to base a sample size calculation from. However, prior neural correlate approaches have been successful with smaller sample sizes. Thus, the full cohort sample size of 31 was deemed satisfactory to allow for at least preliminary effect size calculations and the follow-up sub-group analysis based on injury risk classification was exploratory. We have added a statement to this effect in the limitations:

“As this was a preliminary investigation on a novel fMRI motor task and brain-behavioral correlate, the small sample size is a limitation, though is in line with prior fMRI sample sizes for similar tasks[24,25,91], and will allow for effect size calculation for future larger studies”

Page 6 Lines 139-140: “thereby facilitating a greater correspondence with actual neuromuscular control demands of the DVJ”. I suggest removing this sentence since the speculation may send out an ambiguous message considering the complexity and the real neuromuscular-coordinative demand of a jump-landing maneuver with respect to the bilateral leg press task.

RESPONSE: We agree and revised the sentence to be: 

“The bilateral leg press involved both a concentric press phase, an eccentric loading phase and a range of motion with greater ecological relevance to actual landing than prior methods, though still limited by neuroimaging constraints.”

Page 8 Line 174: How did you measure maximum jump height? Please argue on that

RESPONSE: We are now specific in the manuscript text: 

“Prior to the testing session, maximum counter movement vertical jump height was determined by having them attempt to grab a basketball at its maximum height on custom basketball retractor. Once ball height was raised to a height that ball could not be retrieved the highest successful measure was recorded and used to set the ball target height during the drop vertical jump testing.”

Discussion

Page 13 line 280: Please change “This assertion is support not only by” with “This assertion is supported not only by”

RESEPONSE: Thank you for catching this, revised as requested. 

Page 13 line 281: Please change “but the implicated sensory” with “but also by the implicated sensory”

RESPONSE: Thank you for catching this, revised as requested.

Limitations

Please add the small sample size as a limitation to the generalizability of the study results.

RESPONSE: We agree and have added a sentence to reflect this: 

“As this was a preliminary investigation on a novel fMRI motor task and brain-behavioral correlate, the small sample size is a limitation, though is in line with prior fMRI sample sizes for similar tasks[24,25,91], and will allow for effect size calculation for future larger studies.”

---

## [Decision Letter · Decision Letter 1]

22 Jul 2022

Preliminary Brain-Behavioral Neural Correlates of Anterior Cruciate Ligament Injury Risk Landing Biomechanics Using A Novel Bilateral Leg Press Neuroimaging Paradigm

PONE-D-21-23995R1

Dear Dr. Grooms,

We’re pleased to inform you that your manuscript has been judged scientifically suitable for publication and will be formally accepted for publication once it meets all outstanding technical requirements.

Kind regards,

Riccardo Di Giminiani

Academic Editor

PLOS ONE

Additional Editor Comments (optional):

Reviewers' comments:

Reviewer's Responses to Questions

**Comments to the Author**

1. If the authors have adequately addressed your comments raised in a previous round of review and you feel that this manuscript is now acceptable for publication, you may indicate that here to bypass the “Comments to the Author” section, enter your conflict of interest statement in the “Confidential to Editor” section, and submit your "Accept" recommendation.

Reviewer #1: All comments have been addressed

Reviewer #2: All comments have been addressed

2. Is the manuscript technically sound, and do the data support the conclusions?

Reviewer #1: Yes

Reviewer #2: Yes

3. Has the statistical analysis been performed appropriately and rigorously? 

Reviewer #1: Yes

Reviewer #2: Yes

4. Have the authors made all data underlying the findings in their manuscript fully available?

Reviewer #1: Yes

Reviewer #2: Yes

5. Is the manuscript presented in an intelligible fashion and written in standard English?

Reviewer #1: Yes

Reviewer #2: Yes

6. Review Comments to the Author

Reviewer #1: I have no further comments. I thank the authors for their commitment in answering all the main points I have raised.

Reviewer #2: (No Response)

7. PLOS authors have the option to publish the peer review history of their article (what does this mean?). If published, this will include your full peer review and any attached files.

Reviewer #1: No

Reviewer #2: **Yes: **jacopo emanuele rocchi

---

## [Editor Report · Acceptance letter]

2 Aug 2022

PONE-D-21-23995R1 

Preliminary Brain-Behavioral Neural Correlates of Anterior Cruciate Ligament Injury Risk Landing Biomechanics Using A Novel Bilateral Leg Press Neuroimaging Paradigm 

Dear Dr. Grooms:

I'm pleased to inform you that your manuscript has been deemed suitable for publication in PLOS ONE. Congratulations! Your manuscript is now with our production department. 

Kind regards, 

on behalf of

Prof Riccardo Di Giminiani 

Academic Editor

PLOS ONE